# An Assessment of the Reliability and Validity of the PERMA Well-Being Scale for Adult Undergraduate Students in an Open and Distance Learning Context

**DOI:** 10.3390/ijerph192416886

**Published:** 2022-12-15

**Authors:** Ishmael Magare, Marien Alet Graham, Irma Eloff

**Affiliations:** 1Department of Educational Psychology, University of Pretoria, Pretoria 0002, South Africa; 2Department of Science, Mathematics and Technology Education, University of Pretoria, Pretoria 0002, South Africa

**Keywords:** open and distance learning, well-being, PERMA model, undergraduate students, reliability, validity, confirmatory factor analysis, sub-Saharan Africa

## Abstract

Background: The PERMA well-being scale measures the multidimensionality of well-being in human populations. It highlights positive emotions, engagement, relationships, meaning, and accomplishment. Despite the empirical advancement and evolution of the PERMA scale in different settings, its applicability to open and distance learning (ODL) has not been adequately established among undergraduate students in sub-Saharan Africa. Methodology: Our study examines the theoretical reliability, validity, and five-factor structure of the shortened 35-item version of the PERMA well-being scale as it was adapted in an ODL tertiary institution in Botswana. The PERMA model of well-being and self-determination theory (SDT) served as theoretical frameworks. We evaluated the adapted PERMA scale’s reliability, construct validity, confirmatory factor analysis, and measures of invariance to assess if the data of undergraduate students in an ODL context study fitted the PERMA model of a well-being five-factor structure. We used a multi-stage sampling scheme incorporating a convenience sampling approach where the respondents were invited to voluntarily participate in the study through a WhatsApp group, followed by snowball sampling where we asked the participants to add others to the WhatsApp group during the timeline of the survey; the sample comprised 215 respondents (age: mean = 38.17, standard deviation = 6.472). We collected data from former and active undergraduate B.Ed. (Bachelor of Education) degree students from five regional campuses of the open university through an online survey built into the Qualtrics platform. The Cronbach’s alpha indicated that one item should be removed from the engagement domain. Results: The overall adapted scale retained a 34-item PERMA well-being scale in the particular ODL context. The goodness of fit indices confirmed the five-domain structure with the 34 items. Conclusions: The psychometric properties of the 34-item adapted PERMA well-being scale suggest that it can be a valuable and feasible instrument in ODL in sub-Saharan Africa. Furthermore, the adapted scale can be applied in educational settings moving towards open and distance e-learning forms of delivery.

## 1. Introduction

Well-being is on the global agenda for governments and inter-governmental organisations as part of the Sustainable Development Goals (SDGs) of the United Nations [1,2]. It is articulated explicitly as SDG 3 on “Good Health and Wellbeing”. Well-being correlates with positive outcomes: life expectancy indices, good relationships, productivity, creativity, and meaningful learning [3]. From extensive research and theoretical analysis in combination with conceptualisations of human experiences and interpretations, [4] developed a theoretical model of PERMA well-being that has been used and validated through the years [5,6,7,8,9,10,11]. The model, also referred to as the PERMA framework, is based on five pillars: Positive Emotions (PE), Engagement (EN), Relationships (REL), Meaning (MNG), and Accomplishment (ACC) [12,13,14,15]. The PERMA model is a theory of well-being based on Seligman’s extensive understanding and thinking around various areas of scholarship, combined with his perceptions and interpretations of people’s experiences. 

The PERMA model as an established psychological measurement scale still must be applied across diverse cultural backgrounds, ethnicities, domains, and institutions to have universal reliability and validity [16,17]. The latter authors [17] stress that particular attention needs to be paid to the following Seligman criteria, namely that each of the five core domains of the PERMA theory should (1) contribute to well-being; (2) be pursued for its own sake, not merely to obtain any of the other well-being domains; and (3) be discrete and not dependent on other elements in terms of description and measurement.

The development of the PERMA theory is a product of numerous studies conducted in the field of positive psychology and the quest for the PERMA scale as a global measure of well-being is ongoing [9,12,14,18]. Empirical research in various fields has tested the PERMA scale and its theoretical assumption and has largely confirmed its reliability and five-domain psychometric characteristics [5,7,9,12,14,15,16,17,18,19,20]. Of the latter authors, [15] maintains that positive emotions are crucial to well-being. Positive emotions describe the good feelings that motivate individuals into action and include joy, happiness, cheerfulness, and ecstasy. The feelings contribute to improved human functioning and growth [7], which facilitate expansive reasoning and result in congruent adaptive skills and behaviours. The same authors [7] show that positive emotions associate positively with psychological human functions such as self-concept, attention, creativity, generativity, contentment, arousal, and social integration. Positive emotions relate to objective well-being (hedonic factors), which are contingent on pursuing pleasurable outcomes and escaping unpleasant conditions [21]. Other authors [6] extrapolate the ideation that positive emotions (productive social relations, physical vitality, longevity, success at the workplace, and general coherence in life) correlate with the frequency with which they are experienced and do not occur in the absence of negative emotions and pain. The same authors [6] posit that positive emotions result from recovery from uncomfortable circumstances that warrant practical and objective responses. 

Engagement, a product of flow, refers to the level of absorption in an act to such an extent that one loses count of time. Engagement has its roots in the work of [22] on the concept of “finding flow”, where time seems to “stand still as an individual loses the self with the intense concentration in an activity at hand” and [18] indicates that flow annihilates time with blissful immersion in the present moment. Flow is a state of enjoyment that cuts across cultures, gender, age, disciplines, and occupational contexts [23,24]. Three dimensions of engagement (behavioural, psychological, and cognitive) that define students have been reported [14], while working adults’ engagements revolve around work commitment and the elderly commit to social engagement at the communal level. Other authors [15] maintain that engagement demands skill and effort and leverages the individual’s signature strengths. Engagement is a source of gratification and inspiration that overshadows salaries and other substantial benefits in the workplace. Engagement is a predictor of good grade point averages (GPAs) among college students and promotes positive affect and life satisfaction [9].

Relationships form the third element of the PERMA model of well-being [9,12,13,18]. Humans are social beings and value relationships with their significant others, such as family members, friends, and co-workers, who form part of belonging in a reciprocal, bonded, and mutual relationship [12,14]. The enhancement of well-being often depends on positive relationships and connections with family members and friends. Such networks provide a purpose for existence through sharing triumphs and defeats [25,26,27,28]. Relationship-based well-being is not dependent on others but infers being present and accessible at the most needed time [13]. Relationships thrive on listening and allowing significant others to relate openly and unconditionally to opinions expressed by others.

Meaning constitutes the fourth element in the PERMA model. It involves individuals’ introspective, reflective, and mirroring processes for defining events in their lives concerning the past, the present, and the future [29,30]. Meaning is associated with personal identity and giving rather than taking. Meaningfulness involves integrating the past, the present, and the future [9,17,31,32,33,34]. Human beings use strengths outside personal beneficence in fulfilling the goals they consider significant [7]. A high sense of meaning is associated with life satisfaction, positive affect, and academic achievement in college students [9]. Human beings subscribe to public groups, philanthropism, religious movements, and voluntary work to achieve meaning. These affiliations and “belonging” bring a sense of purpose, hope, fulfilment, and value to their lives [7]. 

Accomplishment has often been linked to a persevering attitude rather than an outcome alone [15]. In this regard, [18] shows that, subjectively, accomplishment implies mastery and competence in living a productive and meaningful life and is a means to happiness. Objectively, it involves achievements attained through individual skills and efforts aligned with specific strategic goals in life. Attainments may be in the form of victories, awards, and other tangible expressions. Achievement usually augments well-being, even if it does not correlate with positive emotions, meaning, and engagement [7]. Research [9,14] shows that achievement is positively associated with good social standing, health, and well-being, and is a good predictor for GPAs among college students.

In conjunction with the PERMA model, the current study was also informed by self-determination theory (SDT). This theory provides conceptual markers for understanding the motivational dynamics and stable psychological and operational behaviour that are reflected in two states of attentiveness, namely mindfulness and interest [35], which were considered relevant in the study.

Our paper examines and extends Seligman’s PERMA theoretical model in that we attempted to test the model’s psychometric properties among students in an ODL university context in Botswana. The study adopted an Afrocentric perspective that recognises contextual dynamics, with B.Ed. degree undergraduate students as participants. Although the PERMA model is an emerging well-being theory in positive psychology, more research is needed to verify its reliability and validity in different global contexts. The COVID-19 pandemic and post-COVID-19 eras experienced a major move towards digital technologies in the dispensation of learning, especially in ODL higher education contexts [36,37,38]. Social and emotional values as well as students’ engagement remain crucial to well-being in a sustainable educational milieu and directly promote quality of life. Digital technologies cause students’ physical presence on campuses and classrooms to be redundant and can support practical learning experiences, connect students to their significant others, and enhance their well-being [39]. 

The objective of the study was to investigate whether the research data fitted the five-domain structure of the PERMA framework and its psychometric properties among undergraduate students in a Botswana ODL context. The empirical investigation of the PERMA framework in ODL institutions and various student populations is growing [40,41]. The current study aimed to contribute to knowledge development with the help of education students from Botswana. First, we assessed whether the PERMA sub-constructs reached the threshold of acceptable reliability coefficients. Second, we evaluated construct validity by establishing the convergent and discriminant validity of the PERMA sub-constructs by running correlations. Finally, we determined whether the data on the ODL undergraduate students fitted the five-factor structure of the PERMA model of well-being. We used reliability indices, construct validity, confirmatory factor analysis (CFA), and measurement invariance (MI) to evaluate the 35 items of the adapted PERMA profile scale. 

## 2. Methods

### 2.1. Participants

The study was conducted at an ODL university from 2020 to 2022 with participants pursuing a B.Ed. degree in primary education. We used a multi-stage sampling scheme incorporating a convenience sampling approach where the respondents were invited to voluntarily participate in the study through a WhatsApp group. The inclusion criteria for the three cohort groups were students who had enrolled for a B.Ed. degree—primary between 2015 and 2017 at the open university and had completed their studies in the subsequent years. This was a non-probability sampling method, as the invitation was posted on a WhatsApp group, leaving it up to the students to participate or not participate. The second stage involved snowball sampling where we asked the participants to add others to the WhatsApp group during the timeline of the survey. The students agreeing to participate in the study followed the link provided to sign the consent form and complete the survey on the Qualtrics platform. The sample comprised 215 respondents (13.5% male and 86.5% female students), which reflected the B.Ed. primary programme population at the open university where the study was conducted. The ratio of male to female students has always been predominately in favour of the female students. The students were distributed across three Southern African countries and one Eastern African country. The respondents’ mean age was 38.23 years (*Md* = 37.00, *SD* = 6.47, *IQR* = 8.00) and the mean (*M*) teaching experience was 12.66 years (*Md* = 12.00, *SD* = 7.17, *IQR* = 8.00), where *Md*, *SD,* and *IQR* stand for the median, standard deviation, and interquartile range, respectively. Note that the *Md* and *IQR* are reported alongside the *M* and *SD* since the data are non-normal (normality tests are discussed later in the Data Analysis section). Note also that the participants in the study had some “teaching experience” as they had a diploma in primary education (DPE) and had enrolled for the B.Ed. degree to upgrade their qualifications. The advanced age profile is typical for undergraduate education students in sub-Saharan Africa, who often obtain teaching qualifications in later life stages. The respondents were spread over ten mother tongue languages, with Setswana spoken by 76.6% and Kalanga by 6.2% of the respondents. The other remaining language groups accounted for less than 4.0% of the respondents. The respondents reported their year of admission to the university in three consecutive years: 2015 (14.0%), 2016 (32.0%), and 2017 (54.0%). The five regional campuses of the university were used to draw the sample. The ethical clearance was obtained from the Faculty of Education Ethics Committee, University of Pretoria, to undertake the study under ref. EDU040/19. 

### 2.2. Measures

We adopted and contextualised the shortened versions of the PERMA framework questionnaires used to measure students’ work-related well-being [14,18] in terms of the adult PERMA profiler. The PERMA profiler is a 35-item scale that includes Positive Emotions (PE), Engagement (EN), Relationships (REL), Meaning (MNG), and Accomplishment (ACC), which are drawn and reduced from several scales measuring eudaimonia and hedonic well-being [12]. The PERMA profiler scale includes the adolescent measure of well-being built on five model structures: Engagement, Perseverance, Optimism, Connectedness, and Happiness (EPOCH). It also includes the Positive and Negative Affect Schedule for Children (PANAS-C), the Satisfaction with Life Scale (SLS), the Children’s Hope Scale (CHS), the Gratitude Questionnaire (GQ), and the Growth Mindset Scale (GMS). The final adapted PERMA profiler scale [12] in the current study was a five-domain sub-scale instrument constituted as follows: PE (13 items); EN (6 items); REL (7 items); MNG (3 items); and ACC (6 items). It scored the construct on PE on a 5-point Likert scale from Never (1); Rarely (2); Sometimes (3); Very often (4); to Always (5). We scored the other four sub-domains (EN, RE, MNG, and ACC) on a 5-point Likert scale from Strongly Disagree (1) to Strongly Agree (5). The scale was tailored for ODL. For instance, the item “In general, how often do you feel joyful?” was rephrased as “How often did you feel joyful as an ODL university student?”; and “To what extent do you generally feel you have a sense of direction in your life?” was rephrased as “Generally I felt I had a sense of direction in my life while at the university”. The sub-domain items used in this study were as follows:
PE1: How often did you feel cheerful at the university?PE2: How often did you feel joyful as an ODL university student?PE3: How often did you feel energetic as a university student?PE4: How often did you feel delighted as part of the university?PE5: How often did you feel proud of being a university student?PE6: How often did you feel fearless in the pursuit of the degree?PE7: How often did you feel calm towards demanding course requirements?PE8: How often did you feel happy as a university student?PE9: How often did you feel excited as a university student?PE10: How often did you feel active in the pursuit of academic requirements?PE11: How often did you feel daring with seemingly challenging academic requirements?PE12: How often did you feel secure in the face of difficult university tasks?PE13: How often did you feel lively in the pursuit of the degree qualification?EN1: When I read or learnt something new, I often lost track of how much time passed.EN2: I got so involved in activities that I forgot about everything else.EN3: When I saw beautiful reading, I enjoyed it so much that I lost track of time.EN4: How often did you feel interested in completing course activities?EN5: I often got completely absorbed in what I was doing.EN6: How often did you feel alert to the course deadlines?REL1: When I enrolled for the degree my university and family relationships were supportive and rewarding.REL2: I actively contributed to the happiness and well-being of other students at the university.REL3: When something good happened to me at the university, I had people in my life that I liked to share the good news with.REL4: I had friends at the university that I cared about.REL5: There were people in my life at the university who cared about me.REL6: When I had a problem at the university, I had someone who was there for me.REL7: I felt that I was loved at the university and beyond.MNG1: I generally felt that what I did in my life, and especially at the university, was valuable and worthwhile.MNG2: I feel that my university life at the university had a purposeMNG3: Generally, I felt I had a sense of direction in my life while at the university.ACC1: I finished whatever I began at the university.ACC2: Once I made a plan to get something done for studies, I stuck to it.ACC3: I was a hard worker.ACC4: I kept at my university work until I was done with it.ACC5: Most of the times, I felt a sense of accomplishment from what I did for my course work.ACC6: In the recent past, I have been pleased about completing something hard to do.

We also included the respondents’ demographic variables of gender, age, nationality, mother tongue, year of admission to the university, teaching experience, regional campus of registration, place where they taught, the post responsibility, education level, and marital status as part of the instrument.

### 2.3. Data Analysis

We used the IBM Statistical Package for the Social Sciences (SPSS) version 26 and the Analysis of Moment Structures (AMOS) version 26 software packages to analyse the data using a 5% significance level. Only two of the biographical variables had missing values, namely age (*n* = 3) and mother language (*n* = 1). None of the Likert-type questions had missing values. For the missing values, pairwise deletion was used over listwise deletion as the latter leads to a smaller sample size and lower statistical power as the entire record is excluded from analysis if a single value is missing [42].

We used the Shapiro–Wilk test to test the normality data. Because the test violated the normality assumption, we used the nonparametric Spearman correlation tests. We used descriptive statistics to report the respondents’ demographic characteristics and conducted Cronbach’s alpha analysis to establish the internal consistency of the five PERMA constructs. Cronbach’s alpha was used as it is the best-known measure of reliability testing; however, we also reported on McDonald’s omega, as it is well-known that Cronbach’s alpha coefficient increases as the number of items on the sub-domain increase and it decreases as the number of items decrease; thus, it is sensitive to the number of items on the sub-domain, whereas McDonald’s omega does not increase or decrease with the number of items on the sub-domain [43]. In testing for the construct validity of the adapted PERMA scale, we used convergent and discriminant validity tests. We conducted a CFA to test whether the data fitted the five-domain PERMA model. For the CFA, the goodness-of-fit (GOF) measures considered were the normed Chi-square (CMIN/DF), the root-mean-square error of approximations (RMSEA), the goodness-of-fit index (GFI), the adjusted goodness-of-fit index (AGFI), the comparative fit index (CFI), the normed fit index (NFI), and the Tucker–Lewis index (TLI).

We finally used the MI to determine the factor structure of the data against the respondents’ ages. All the statistical tests met the minimum sample size requirements. For normality, recommendations are that the Kolmogorov–Smirnov (K-S) test should be used for samples of size 50 and greater and that the Shapiro–Wilk should be used when the sample size is less than 50; however, the Shapiro–Wilk test provides greater statistical power than the K-S test regardless of the sample size. Researchers (e.g., [44]) maintain that it is the best choice for testing normality for any sample size. For correlation (for the correlation analysis and for establishing validity) using G*Power software [45], the minimum required sample size (*n*_min_) for a small (0.1), medium (0.3), and large (0.5) effect size [46] were 782, 84, and 29, respectively. However, many researchers suggest that it is unnecessary to obtain the minimum sample size required to detect small effect sizes, as finding a statistically significant result for a small effect may have statistical significance (*p* < 0.05) but not real-world or practical significance [47,48]. Thus, ignoring small effect size, a *n*_min_ of 84 and 29 are required for medium and large effect size, respectively. For Cronbach’s alpha, [49] provided a formula for calculating the *n*_min_ which, for the current study, provided *n_min_* equal to 31. In this calculation, the construct with the fewest items (=3) was used, as the fewer items in a construct, the larger the sample size required. For McDonald’s omega, since it is based on parameter estimates (i.e., estimates of factor loadings and factor variances) that are derived for the CFA model, the sample size recommendations for CFA were followed [50]. For CFA, the *n*_min_ needed has been disputed in the literature for decades [51], with recommendations based on a constant value (varying from 100 to 500) or on a minimum number of observations per variable. The recent research indicates that the ratio of observations to variables should be at least two to one and preferably five to one. Using a 5:1 ratio, and given that the questionnaire has 35 items, the *n*_min_ required for the study was 175. 

### 2.4. Quality Criteria

For the quality criteria, reliability and validity were considered. The reliability of an instrument should be considered before considering its validity, since an instrument can be reliable but not valid. However, a measure cannot be valid unless it is reliable. The COSMIN [52], which is a detailed checklist about design requirements (e.g., how the reliability and validity of an instrument were established), was considered in this study. Due to the need for conciseness, not all the details are provided here but only the main ones.

Concerning the reliability of the instrument, we measured the internal consistency of the five PERMA constructs through Cronbach’s alpha analysis with Cronbach’s alpha coefficients being acceptable when above 0.70. The reliability estimates for all five of the PERMA sub-domains yielded acceptable alpha indices (PE = 0.84, REL = 0.82, MNG = 0.87, ACC = 0.73), except for the EN sub-domain when all six items were considered. When EN4 (“I kept at my university work until I was done with it”) was included, α = 0.68; however, with EN4 excluded, the internal consistency of the EN scale was established (α = 0.73; *n* = 5 items). Acceptable values were obtained (PE = 0.84, EN = 0.76, REL = 0.82, MNG = 0.87, ACC = 0.72). Note that EN4 had to be dropped for the calculation of the McDonald’s omega also, as SPSS provides the error message “omega cannot be estimated due to negative or zero item covariance” when EN4 was included in the EN sub-domain. This further justified EN4′s exclusion from the analysis. 

Concerning the validity of the instrument, we tested the adapted PERMA profiler scale for construct validity through convergent and discriminant validity tests. Convergent validity means that items belonging to the same construct/factor correlate more strongly than items belonging to different constructs/factors. On the other hand, discriminant validity means that items belonging to various constructs/factors correlate less strongly than items belonging to the same construct/factor. Not all the correlations are shown for reasons of conciseness and the results are summarised as discussed below. It should be noted that EN4 was not included when establishing validity, as one first must establish reliability before one can establish validity. As reliability could be established only when EN4 was excluded, the validity was also considered with the exclusion of EN4. Regarding convergent validity for the PE, EN, REL, MEAN, and ACC constructs/factors, the correlations reached values as high as 0.761, 0.616, 0.682, 0.695, and 0.516, respectively. Regarding discriminant validity, the maximum values for the correlations were much lower, with the highest correlation between items belonging to different constructs/scales equalling 0.487. These results indicate that validity was established.

## 3. Results

### 3.1. Descriptive Statistics

Table 1 shows the descriptive statistics for the individual items of the sub-domains and for the overall sub-domains. Note that, for the overall sub-domains, the values of the individual items were averaged. Additionally, note that this was performed for the theoretical PERMA model where all the items were included. Later on, when establishing reliability of the sub-domains, it will be seen that EN4 was dropped from the EN sub-domain.

We measured all the items on all five domains on a five-point Likert scale. Higher scores (above the midpoint of three) indicated relative agreement with each item. Conversely, lower scores (below the midpoint of three) indicated disagreement with each of the items on the sub-domains. Table 1 presents the descriptive statistics (*M*, *SD*, *Md*, and *IQR*) for each of the 35 items of the PERMA framework as self-reported by the undergraduate students (*n* = 215) in the ODL university context. For all items except for EN2, the *M* and *Md* were above the midpoint of 3, indicating that the respondents agreed with the statements. For EN2 (“I got so involved in activities that I forgot about everything else”), the *M* and *Md* were below the midpoint of 3, indicating that the respondents disagreed with this statement. For the overall scores, the MNG sub-domain had the highest scores, indicating the importance of meaning (e.g., sense of purpose in life, feeling that life is worth living, and connecting to something greater than the self), whereas EN had the lowest scores. However, it will be shown later that EN4 was removed from the EN sub-domain to establish its reliability. With EN4 removed, the statistics were slightly lower with a mean of 3.59 (*SD* = 0.74) and a median of 3.60 (*IQR* = 1.20). Thus, the EN sub-domain, after establishing its reliability in the next sections, had the lowest scores, indicating that engagement (e.g., being absorbed, interested, and involved) might have presented a challenge within this population. 

### 3.2. Confirmatory Factor Analysis

We conducted a CFA to confirm the underlying factor structure (see Table 2). It should be noted that EN4 was included in Model A (the complete theoretical model); however, the CFA results agreed with those of Cronbach’s alpha to remove EN4. See Model B (the final theoretical model). 

In the current study, we investigated whether the five-domain structure of the PERMA model of well-being could be empirically derived from our data on undergraduate students in an ODL context in sub-Saharan Africa. We evaluated the model fit (Table 2) value for the complete initial theoretical model (A) based on the sample population of the study (*n* =215) on 35 items against the GOF indices thresholds (CMIN/DF = 1.944, RMSEA = 0.066, GFI = 0.762, AGFI = 0.727, CFI = 0.816, and TLI = 0.801). The RMSEA is primarily a population-based index that cannot be affected by sample sizes [12] and other additional GOF statistics. 

We extracted the thresholds model fit indices from [53], who adapted the thresholds for model fit indices from [54,55,56], collated their recommendations, and presented them in their manuscript [53]. From Table 2, it can be seen that these recommendations differ from conventional recommendations in the sense that, using the CFI for example, conventional recommendations typically state that CFI ≥ 0.9 [57], whereas [53] states that a value of zero for CFI indicates “no fit”, while values of one indicate “perfect fit”, so the higher the CFI value, the better. These conventional cut-off values of, for example, CFI ≥ 0.9, have been debated in the literature as being too strict; see for example [58]. These authors argue that smaller RMSEA and larger CFI and TLI (to mention just a few thresholds) indicate better fit and that one should not depend solely on the conventional cut-off values recommended decades ago by, for example, [57]. From Table 2, the value for the complete initial theoretical model (A) column, it can be seen that all the model fit indices looked good since the CMIN/DF were between 1 and 2, the RMSEA was less than 0.08, and the rest of the model fit indices were relatively close to one. However, the NFI was below 0.7 and we had to adjust the model to improve it. When running the reliability analysis, we found that the item EN4 “How often did you feel interested in completing course activities?” did not fit well with the EN construct. We therefore re-ran the CFA with EN4 excluded.

The GOF statistics for the complete final theoretical model (B) indicate that the model improved because the NFI increased from 0.687 to 0.714 and all the other model fit indices (CMIN/DF =1.822, RMSEA = 0.062, GFI = 0.781, AGFI = 0.748, CFI = 0.844, NFI = 0.714, and TLI = 0.831) were acceptable (Table 2). The five-domain theoretical structure of the PERMA model was supported and was a necessary step before developing the structural model. 

The next step was to test for measurement invariance [59], which is also referred to as multigroup CFA (MGCFA) [44]. MGCFA is an extension of typical CFA with the difference that instead of fitting a single model to the entire data set, we split the data into two (or more) groups and determined the model fit for each group. This procedure allowed us to explore whether the respondents from different groups similarly conceptualised the constructs. We investigated all demographic variables to decide which groups to compare in the MGCFA and we chose to use age. The reason for the choice of age was that, for example, when considering nationality, 196 respondents indicated Motswana, 14 indicated Zimbabwean, 3 indicated Zambian, and 2 indicated Kenyan. Since the sample size would not allow a CFA to be run according to nationality (e.g., a CFA cannot be performed on data sets (groups) of sizes 2, 3, and 14), we did not consider nationality. This was the problem with most of the demographic variables.

In many cases, the demographic variables had sparse data; however, in terms of the age variable, we could create two groups using the continuous variable age. We used the median (age 37) to create the two groups. We classified respondents 37 years or younger as younger respondents and respondents older than 37 as older. We used these two groups for testing measurement invariance as follows: the younger group (51.0%) and the older group (47.0%), respectively. Some of the respondents (2.0%) did not indicate their age and we therefore excluded them from the measurement invariance tests. 

The model fit indices most often reported for measurement invariance testing are the Chi-square and its corresponding degrees of freedom (df), the RMSEA, the standardised root mean-square residual (SRMR), the CFI, and the TLI. Although the Chi-square test is typically reported in measurement invariance testing, it is difficult to interpret as the general rule is simply the smaller, the better. Therefore, interpreting the normed Chi-square is easier (as it must ideally be between 1 and 2) (Table 3). Note that the result will be the same regardless of whether the Chi-square or the normed Chi-square is provided, as the latter is simply the Chi-square test statistic divided by the df. There are four levels of measurement invariance. Each of these levels builds upon the previous level by introducing additional equality constraints on model parameters to achieve more robust forms of invariance [59]. The levels are configural invariance, metric invariance, scalar invariance, and strict invariance. 

#### 3.2.1. Configural Invariance (Configural Model)—Equal Form

The configural invariance test allowed us to examine whether the overall domain structure fitted the younger and older age groups well. In other words, were the number of domains and the pattern of domain-indicator relationships the same for the younger and older age groups. To test configural invariance, the model fitted the younger and older age groups, leaving all domain loadings and item intercepts free to vary for each age group. From Table 3, it can be seen that the configural invariance held across the two age groups as the model fit was acceptable. This means that the younger and older respondents conceptualised the constructs similarly.

#### 3.2.2. Metric Invariance (Metric Model)—Equal Loadings

For metric invariance, the domain loadings were constrained to be equivalent across groups, while the item intercepts were free to vary for each age group. When assessing metric invariance, the metric-model fit was compared to the configural-model fit and from Table 3 it can be seen that the SRMR moved from a value below the cut-off of 0.09 to a value above it. This suggests that there was no domain loading invariance. This indicates that the domain loadings differed significantly between the younger and older respondents. It is worth noting that the other GOF measures, besides the SRMR, demonstrated a good model fit, which led us to recommend that a follow-up study be conducted with a larger sample size to explore further the issue of metric invariance within undergraduate students in an ODL context. 

#### 3.2.3. Scalar Invariance (Scalar Model)—Equal Intercepts

Scalar invariance builds upon metric invariance by requiring that the item intercepts also be constrained to be the same across younger and older age groups. The scalar invariance is explained [60] as follows: “Establishing scalar invariance indicates that observed scores are related to the latent scores; that is, individuals who have the same score on the latent construct would obtain the same score on the observed variable regardless of their group membership” (p. 115). When assessing scalar invariance in the current study, the scalar-model fit was compared to the metric-model fit. From Table 3, it can be seen that the SRMR remained a problem as it was slightly greater than the cut-off value of 0.09, indicating no intercept invariance. This indicates that the intercepts differed significantly between the younger and older respondents. It is worth noting that the other GOF measures, other than the SRMR, indicated a good model fit, which led us to recommend that a follow-up study be conducted with a larger sample size to further explore the scalar invariance within the undergraduate students in an ODL context. 

#### 3.2.4. Strict Invariance (Strict Model)—Equal Residual Variances

Strict invariance is the most stringent level of measurement invariance testing. Here, the residual variance of the observed scores, not accounted for by the constructs/domains, are considered. So, when testing strict invariance in the current study, we evaluated whether the residual error was equivalent across the younger and older respondents. From Table 3, it can be seen that the GOF measures for the strict model did not differ significantly from the scalar model, with the SRMR value continuing to be the only GOF measure indicating a problem. This indicates that the residual variance of the observed scores, not accounted for by the domains, differed significantly between the younger and older respondents. Experts (see, for example, [59]) agree that it is unreasonable to expect equality in residual variances across groups, so it was not surprising that the GOF measures for the strict model were the worst (although not significantly different from the rest, it still had the worst GOF measures). In conclusion, from Table 3, it can be seen that the configural invariance held across the two age groups as all GOF measures fell within acceptable ranges. However, for metric invariance, scalar invariance, and strict invariance, all the GOF measures showed good model fit, except for the SRMR.

## 4. Discussion

The current study examined the reliability and construct validity of the five-domain structure of the PERMA framework and its psychometric properties among undergraduate students in an ODL context in a sub-Saharan African country, Botswana. To our knowledge, this is the first study of its kind in an ODL environment in the country. The well-being of undergraduate students in such an environment is critical in the post-COVID migration to digital systems. The study evaluated the shortened version of the PERMA scale through measures of internal consistency, construct validity, CFA, and measurement invariance models. We investigated the internal consistency of the shortened 35-item PERMA model through Cronbach’s alpha coefficients. We excluded Item 4 (EN4: “How often did you feel interested in completing course activities?”) of the engagement domain resulting in a 34-item scale due to its questionable Cronbach’s coefficient value threshold in respect of the other items. However, its exclusion did not compromise the construct validity of the overall engagement domain. The five domains of the PERMA framework measured through the 34 items demonstrated acceptable (above 0.7) internal consistency values through the Cronbach alpha reliability indices. The PERMA scale internal consistency shows its consistency with the original versions of the measure and its reliability and utility across cultures and ODL involving undergraduate students. The results corroborate the results of the 11 studies conducted for the PERMA Profiler [12] and another study (*n* = 516) conducted in Adelaide, Australia [14].

In addition, the construct validity established in the current study through the convergent and discriminant validity of the five subscales provides evidence of the multidimensionality ability of the constructs to measure different domains (PERMA) of undergraduate well-being in an ODL context. The results of this study corroborate previous research with student veterans from the Army, the Air Force, the Marine Corps, the Navy and the Coast Guard in the US [10] (p. 27), which illustrated the PERMA Profiler as a “multidimensional scale with good reliability and acceptable levels of convergent, divergent, and criterion-related validity”. In the current study, it was important to indicate that the positive emotions subsumed in the 13-item scale reflected a conglomeration of positive-to-negative feelings, high-to-low emotional dispositions, vigour, zeal, grit, and resiliency [3]. It is equally notable that the remaining domains also cut through many domains of well-being and interwove with the self-determination theory (SDT) covering human psychological needs of autonomy, competence, and relatedness. We hypothesised that the PERMA well-being and SDT theories were inseparable and that their interlacing indicated the significance of motivational drives in human flourishing [61,62]. The correlations evidenced each subdomain’s independence by producing low positive correlations against each other and remaining statistically significant besides meaning and engagement [9]. The results support the PERMA model as a multidimensional construct for measuring well-being [12].

The confirmatory factor analysis that tested the model fit indices for the 34-item scale supported the five-domain model and earlier empirical evidence derived from a range of well-being domains [5,9,14,15,17,18,19]. The five constructs structure: positive emotions, engagement, relationships, meaning, and accomplishment are conspicuous in most of the literature surveyed that deals with psychological well-being. Interestingly, in its early stages of development and validation, the shortened, adapted version of the PERMA scale retained its psychometric properties across the cultural, racial, and geographical divide, including ODL in sub-Saharan Africa. In this regard, a study with the Greek version of the PERMA profiler demonstrated “acceptable internal consistency and test–retest reliability for the overall well-being items and almost all well-being components” [63] (p. 3030). Regarding the biographical variables, the invariance test on the two age groups revealed that the PERMA structural model had conceptual similarities. Simply put, there were no differences in how the two age groups of students (young and old) conceptualised the PERMA constructs.

Our study examined the shortened, adapted PERMA scale and was able to situate it in an ODL context in Botswana, which in our view represents an expansion of PERMA research and advancement. We also acknowledge that the cross-sectional survey we adopted in the study had limits in comparing it to longitudinal studies, which use different time horizons and subpopulation groups. We also recognise the limitations of the population and sample, which might not necessarily be representative of a university-wide undergraduate student population. However, the fact that higher education is opening up for previously marginalised groups in sub-Saharan Africa should be acknowledged. The adult populations who were initially excluded from higher education opportunities now have options to access tertiary education. Additional studies on the formulation and inclusion/exclusion of Item 4 in African populations are also recommended. Due to the inclusion criteria for students who among other things had to have *successfully* completed their four-year degrees, a further limitation was the time lag between the data collection and period of study. Despite these limitations, we believe that our study contributed valuable insights into the adaptability, stability, reliability, and construct validity of the adapted PERMA scale in an Afro-centric context.

The study has practical as well as theoretical implications in the field. Practically, the study suggests the utility of an adjusted PERMA profiler in tertiary environments in sub-Saharan Africa. Theoretically, the study indicates the need for fine-grained item analysis of the PERMA profiler in the African context. In terms of well-being, the study points to the need for further research on the construct of engagement for students in ODL. 

## 5. Conclusions

The study results showed that the adapted PERMA scale of well-being was a plausible instrument for ODL in sub-Saharan Africa. We were able to retain the PERMA scale’s global five-domain PERMA structure with the present data. The PERMA scale, though still in its developmental stages, is relatively stable and is a valuable addition to the field of instrument development and understanding of PERMA well-being in positive psychology. It also contributes to the development of well-being theory. The current study extended previous empirical research and ushered ODL into the debate and topography of the PERMA model of well-being. Equally notable, in our study, was that the adapted PERMA scale showed its ability to cut across nationalities, age, and other demographic factors with some degree of stability. However, future studies should consider a larger sample size to replicate our research and further probe why “EN4” could not yield a reliable score as part of the Engagement sub-scale. 

## Figures and Tables

**Table 1 ijerph-19-16886-t001:** Descriptive statistics for individual items of the sub-domains and for the overall sub-domains.

PE	*M* (*SD*) *Md* (*IQR*)	EN	*M* (*SD*) *Md* (*IQR*)	REL	*M* (*SD*) *Md* (*IQR*)	MNG	*M* (*SD*) *Md* (*IQR*)	ACC	*M* (*SD*) *Md* (*IQR*)
PE1	3.78 (0.84) 4.00 (1.00)	EN1	3.40 (1.22) 4.00 (1.00)	REL1	4.52 (0.71) 5.00 (1.00)	MNG1	4.62 (0.62) 5.00 (1.00)	ACC1	4.49 (0.83) 5.00 (1.00)
PE2	3.86 (0.89) 4.00 (2.00)	EN2	2.96 (1.24) 3.00 (2.00)	REL2	4.39 (0.61) 4.00 (1.00)	MNG2	4.60 (0.59) 5.00 (1.00)	ACC2	4.28 (0.86) 4.00 (1.00)
PE3	3.90 (0.84) 4.00 (2.00)	EN3	3.28 (1.21) 4.00 (2.00)	REL3	4.49 (0.63) 5.00 (1.00)	MNG3	4.58 (0.61) 5.00 (1.00)	ACC3	4.52 (0.70) 5.00 (1.00)
PE4	4.02 (0.91) 4.00 (2.00)	EN4	4.52 (0.65) 5.00 (1.00)	REL4	4.62 (0.52) 5.00 (1.00)			ACC4	4.47 (0.69) 5.00 (1.00)
PE5	4.5 (0.78) 5.00 (1.00)	EN5	3.87 (0.90) 4.00 (2.00)	REL5	4.45 (0.65) 5.00 (1.00)			ACC5	4.45 (0.66) 5.00 (1.00)
PE6	3.45 (1.23) 4.00 (1.00)	EN6	4.46 (0.69) 5.00 (1.00)	REL6	4.32 (0.80) 4.00 (1.00)			ACC6	4.47 (0.74) 5.00 (1.00)
PE7	3.52 (1.01) 3.00 (1.00)			REL7	3.87 (0.87) 4.00 (1.00)				
PE8	4.23 (0.88) 4.00 (1.00)								
PE9	4.19 (0.89) 4.00 (1.00)								
PE10	4.24 (0.79) 4.00 (1.00)								
PE11	3.46 (0.82) 3.00 (1.00)								
PE12	3.47 (0.91) 3.00 (1.00)								
PE13	4.26 (0.76) 4.00 (1.00)								
Overall	3.92 (0.52) 3.92 (0.77)		3.75 (0.63) 3.83 (1.00)		4.38 (0.48) 4.43 (0.86)		4.60 (0.54) 5.00 (0.67)		4.44 (0.48) 4.50 (0.67)

**Table 2 ijerph-19-16886-t002:** Results of the CFA.

Results of GOF Statistic	CMIN/DF	RMSEA	GFI	AGFI	CFI	NFI	TLI
Recommended level or range	<5 (preferably between 1 and 2)	<0.10 (preferably < 0.08)	0 (no fit)–1 (perfect fit)				
Value for the complete initial theoretical model (A)	1.944	0.066	0.762	0.727	0.816	0.687	0.801
Value for the complete final theoretical model (B)	1.822	0.062	0.781	0.748	0.844	0.714	0.831

**Table 3 ijerph-19-16886-t003:** Results of the measurement invariance tests

Statistic	CMIN/DF	RMSEA	SRMR	CFI	TLI
Recommended level or range	<5 (preferably between 1 and 2)	<0.10 (preferably < 0.08)	< 0.09	0 (no fit)–1 (perfect fit)	0 (no fit)–1 (perfect fit)
Configural	1.625	0.055	0.087	0.783	0.764
Metric	1.628	0.055	0.111	0.775	0.763
Scalar	1.623	0.054	0.117	0.774	0.765
Strict	1.663	0.056	0.117	0.752	0.75

## Data Availability

Data will not be made available for confidentiality reasons.

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
