# Peer review of "An Assessment of the Reliability and Validity of the PERMA Well-Being Scale for Adult Undergraduate Students in an Open and Distance Learning Context"

_ijerph, 2022, doi:10.3390/ijerph192416886_

Round 1
Reviewer 1 Report
ID: ijerph-2032543
Title: An assessment of the reliability and validity of the PERMA well-being scale on undergraduate students in an ODL context.
Thank you for providing a chance to review this manuscript.
Comment: major revision.
Title
Please do not include abbreviations in the title.
Abstract
Your summary is long, chaotic, and illogical. We generally divide the abstract into four parts: background/purpose, methodology, results, and conclusions. Keep the abstract down to leave the most important content, please.
Methods
Line 155, page 4: “The sample comprised 13.5% male and 86.5% female students” ------ Does such a large difference in sample sizes between men and women affect the confidence of the findings?
Line 156-165, page 3-4: This paragraph should appear in the results. Such a detailed description of the sample is not required in the method section.
Line 148-168, page 4-5: How is the theoretical sample size calculated for this study? What is the sampling method? Where are the inclusion and exclusion criteria?
Line 169-189, page 4: In general, the introduction to a scale should include the following parts: scale items, scale dimensions, content of the measure, scoring method, total score range, trends represented by high or low scores, and psychometric properties. I suggest that you should rewrite it according to this section.
Line 196, page 4: In addition to Cronbach's α, please report McDonald’s ω as well.
Line 190-200, page 4: Are there missing values? What percentage of missing values are there? How is it handled?
Follow the COSMIN guidelines, please.
Results
Line 220-222, page 6: Why did you think of removing item 4? I didn't see any foreshadowing at the front of the article.
Line 271-276, page 7: The CFI, TLI, and NFI are all too low, and you mention that are acceptable. What about the corresponding references?
1) I don't know if you understand the difference between the method, results, and the discussions, please don't mix the methods and discussions in the results!!! 2) The results section is too long, you just need a brief description of the chart.
Discussion
1) Was there a previous study mentioning the deletion of item 4? The reliability, validity, and measurement invariance of item 4 removed from this study are still suboptimal, and I don't think you have discussed the results in depth enough. 2) What are the future directions?
Table
Table 1: The content of this table still needs to be considered, and I think it can show richer content.
Table 2-3: The header says recommended levels, but you seem to be listing the range of the statistic itself. In addition, I recommend swapping rows and columns to make the table look better.
Table: 1) Any abbreviations that appear in the form should be written in full in the notes; 2) Why do some tables omit zeros and some don't? Please change to a uniform format.
The most critical problem is that the results of this study are not good and do not seem to have reached the cut-off value. Moreover, distinguish what should be written in each section. Reading more papers from the TOP journals, to learn the formats, expressions, and of great importance—logic, might help a lot before revising. Last, finding a native English speaker to improve the writing can considerably improve the quality.
Thank you and my best,
Your reviewer
Author Response
See uploaded document with full responses to all reviewer comments.

Reviewer 2 Report
The reviewed paper presents an attempt to validate the PERMA scale in the context of distance learning. The idea is interesting and deserves appreciation. However, I am quite critical of the results themselves. One of my main concerns is that the authors measured the school well-being construct without an explanation of why distance learning may affect students’ well-being and a relevant literature review. Also, the mean age of the participants (M= 38.23 years) suggests not a common age for the student sample. Secondly, we actually do not know when the study was conducted and little is written about the way of developing tool items. Therefore the paper needs an in-depth re-editing and review before publishing.
More precise comments are below.
The abstract section is too long and should be shortened
The authors automatically deleted the names of the authors without consideration and because of that the meaning of some sentences is hard to understand e.g. (…) “Good Health and Wellbeing”. [3]
indicate that”(…); „From extensive research and theoretical analysis in combination with conceptualizations of human experiences and interpretations, [4] developed a theoretical model of PERMA well-being which have been used and validated throughout the years” etc. including bizarre situations where the sentence begins with a number “[17] stress that attention (…); “[15] posited”; “[7] indicate” or ends with it instead of the name of the author e.g. “according to [9]” and so on throughout the entire paper, etc. – the entire paper must be corrected in order to eliminate such mistakes
In the introduction section, the authors start with the structure of the PERMA model, and only after it indicates the requirements of the Seligman theory and the development and sources of the theory, in my opinion, the order should be the opposite
One of the main concerns in this paper is that in fact, the introduction does not justify why the well-being construct (in its structure) proposed in the PERMA model would change due to distance learning. The authors only presented the individual dimensions of the model in the introduction and in 1-2 sentences pointed to the issue of Covid 19 pandemic as an important factor. Moreover, engagement in Seligman’s theory is not the same as study engagement, which was suggested by the sentence
I also have doubts about the sentence that “Currently, there is no evidence of empirical investigation of the PERMA framework in ODL institutions (…) as there is a meta-theoretical analysis made by Lou and Xu (2022), Moreover, some authors did empirical studies in this field e.g. Morgan and Simons (2021) or Mustika et al. (2021)
Therefore the originality of the paper as well as the extent of the existing knowledge is rather poor.
The authors suggested that “The study adopts an Afrocentric perspective with B.Ed. Degree undergraduate students as participants” however did not point out what it means.
Study sample section
Surprisingly, there is no information about the number of participants in this section. The expression “teaching experience” is not clear as you stated that participants were not teachers but undergraduate students – this must be explained
Study procedure: there is no information about the time of data collection. The dates 2015-2017 years are misleading, please insert the precise date of the conducted study. Also, the information about the respondents' year of admission to the university is not necessary – more useful would be the current year of studying
Measures
The way of developing the tool is incomprehensible to me. There is no information on why these methods were selected, on what criteria items from listed scales were selected, but also what selection criteria were used, e.g. why gratitude? why children’s scales were used Children hope or Positive and Negative Affect Schedule for Children especially as you examined adult people?
Results
Tables are cut, and no abbreviations are explained
In table 1 the authors refer to descriptive statistics of 34 items, however, there is no explanation of how the midpoint was evaluated
Validity section
The changes in the scale items should be presented before the result section, with information on how they were developed.
The division sample into two groups by age should be justified not by “problems with demographic characteristics of the sample” but by a literature review that indicates such characteristics as important for well-being structure.
It is not clear to me what you mean by the student veterans term, and why it was used.
It is also not clear to me why you have mentioned SDT theory as you did not present its relation to the PERMA model in the introduction – it only appears in the abstract –which is not relevant to your study aim and in the discussion (two sentences).
The discussion section should be enriched by theoretical approaches to school well-being.
Add study limitations subtitle, and try to develop this part
I also suggest indicating the importance of the paper and the practical and theoretical implications
Author Response

(The authors gave the same response as above.)

Round 2
Reviewer 1 Report
ID: ijerph-2032543
Title: An assessment of the reliability and validity of the PERMA well-being scale on undergraduate students in an ODL context.
I appreciate your efforts to improve the manuscript and to respond to the comments made in the first review process. However, there is still one minor issue that needs to be addressed:
Title
Please do not include abbreviations in the title.
Abstract
Line 21, page 1: “B Ed degree” ------ Any abbreviations that appear must be preceded by the full name.
Line29, page 1: “well-being”? “wellbeing”?
Author Response
For the point that English language and style should be checked, the article has been sent to a native professional language editor; the language editing certificate is attached here.
Title
Please do not include abbreviations in the title.
ODL in the title has been written out in full.
Abstract
Line 21, page 1: “B Ed degree” ------ Any abbreviations that appear must be preceded by the full name.
It is now written out in full: "B.Ed. (Bachelor of Education)"
Line29, page 1: “well-being”? “wellbeing”?
"Well-being" is used throughout, unless it's a direct quote or in the title of an article in the Reference List.

Reviewer 2 Report
The authors have introduced minimal corrections that are sufficient for the text to be consider for publishing However, I note that the text would gain more value if the introduction was improved to a greater extent.I also suggest clear indications (mostly in the title and abstract) that the participants are adults.
Currently, the title suggests a group of undergraduate students. Especially as the article is published in the adolescent section It is worth emphasizing the cultural context in the introduction - 1 - 2 sentences on this subject are not enough in my opinion.
Author Response
Under the dropdown list of the "English language and style" it was requested that the English language and style be improved.
The article has been edited by a professional language editor that is native in English, and the language editing certificate has been uploaded here in the reviewer centre.
- The authors have introduced minimal corrections that are sufficient for the text to be consider for publishing However, I note that the text would gain more value if the introduction was improved to a greater extent:
- I also suggest clear indications (mostly in the title and abstract) that the participants are adults.
- Currently, the title suggests a group of undergraduate students. Especially as the article is published in the adolescent section, it is worth emphasizing the cultural context in the introduction - 1 - 2 sentences on this subject are not enough in my opinion.
Please note that all language editing changes have been indicated using track changes only, but to address points 2 and 3 above, both track changes and the use of yellow highlight have been incorporated into the manuscript itself.
